# Keeping 21st Century Paleontology Grounded: Quantitative Genetic Analyses and Ancestral State Reconstruction Re-Emphasize the Essentiality of Fossils

**DOI:** 10.3390/biology11081218

**Published:** 2022-08-13

**Authors:** Tesla A. Monson, Marianne F. Brasil, Michael C. Mahaney, Christopher A. Schmitt, Catherine E. Taylor, Leslea J. Hlusko

**Affiliations:** 1Department of Anthropology, Western Washington University, 516 High Street, Bellingham, WA 98225, USA; 2Berkeley Geochronology Center, 2455 Ridge Road, Berkeley, CA 94709, USA; 3Human Evolution Research Center, Valley Life Sciences Building, University of California Berkeley, MC-3140, Berkeley, CA 94720, USA; 4Department of Human Genetics, South Texas Diabetes and Obesity Institute, University of Texas Rio Grande Valley School of Medicine, Brownsville, TX 78520, USA; 5Department of Anthropology, Boston University, 232 Bay State Road, Boston, MA 02115, USA; 6National Center for Research on Human Evolution (CENIEH), Paseo Sierra de Atapuerca 3, 09002 Burgos, Spain

**Keywords:** primates, Cercopithecidae, monkeys, genotype:phenotype mapping, evolution, dentition, phylogeny

## Abstract

**Simple Summary:**

Over the last two decades of biological research, our understanding of how genes determine dental development and variation has expanded greatly. Here, we explore how this new knowledge can be applied to the fossil record of cercopithecid monkeys. We compare a traditional paleontological method for assessing dental size variation with measurement approaches derived from quantitative genetics and developmental biology. We find that these new methods for assessing dental variation provide novel insight to the evolution of the cercopithecid monkey dentition, different from the insight provided by traditional size measurements. When we explore the variation of these traits in the cercopithecid fossil record, we find that the variation is outside the range predicted based on extant variation alone. Our 21st century biological approach to paleontology reveals that we have even more to learn from fossils than previously recognized.

**Abstract:**

Advances in genetics and developmental biology are revealing the relationship between genotype and dental phenotype (G:P), providing new approaches for how paleontologists assess dental variation in the fossil record. Our aim was to understand how the method of trait definition influences the ability to reconstruct phylogenetic relationships and evolutionary history in the Cercopithecidae, the Linnaean Family of monkeys currently living in Africa and Asia. We compared the two-dimensional assessment of molar size (calculated as the mesiodistal length of the crown multiplied by the buccolingual breadth) to a trait that reflects developmental influences on molar development (the inhibitory cascade, IC) and two traits that reflect the genetic architecture of postcanine tooth size variation (defined through quantitative genetic analyses: MMC and PMM). All traits were significantly influenced by the additive effects of genes and had similarly high heritability estimates. The proportion of covariate effects was greater for two-dimensional size compared to the G:P-defined traits. IC and MMC both showed evidence of selection, suggesting that they result from the same genetic architecture. When compared to the fossil record, Ancestral State Reconstruction using extant taxa consistently underestimated MMC and PMM values, highlighting the necessity of fossil data for understanding evolutionary patterns in these traits. Given that G:P-defined dental traits may provide insight to biological mechanisms that reach far beyond the dentition, this new approach to fossil morphology has the potential to open an entirely new window onto extinct paleobiologies. Without the fossil record, we would not be able to grasp the full range of variation in those biological mechanisms that have existed throughout evolution.

## 1. Introduction

The most essential, core moment in paleontology is when someone notices a fossil as something other than a rock and collects it for scientific study. This event is often just a person walking across the landscape, scanning the ground for evidence of past life. While this simple act has been fundamentally the same for generations of paleontologists, the lead-up to that moment and the science that follows have evolved dramatically. The technological advances that have taken us from landline telephones to smartphones have similarly altered how the science of paleontology is conducted. We can see this in the way scientists discover fossil sites. Where fossiliferous sediments were once identified mostly by happenstance, aerial photography, then satellite imagery, and now remote sensing are common tools for field paleontologists [1,2,3]. As well, our protocols for the collection, inventory, and organization of fossils now rely on fine resolution GIS [4] and remote access to the internet [5].

The laboratory side of the science is also remarkably different from 20th century paleontology. Fossils are now imaged by laser scanners as well as through photography [6,7]. Quantification of those scanned surfaces can be performed in three-dimensions with thousands of points, opening the door for new analytical approaches to morphological variation [8,9] and enabling the digital reconstruction of crushed fossils [10]. With the application of computed tomography (CT), paleontologists can more readily study internal bony structures [11,12], giving them the ability to reconstruct soft-tissue anatomies [13,14]. CT scans have become an essential tool in the description of new fossils [15]. With a synchrotron, we can even see fossilized histology without mechanically damaging specimens [16,17]. Advances in geochemistry provide new insight into the evolution of dietary niches [18,19,20,21] and life history [22], not to mention the ability to geologically date fossils [23]. As well, of course, advances in artificial intelligence and machine learning have forever changed taphonomy [24,25], approaches to fieldwork [26,27], and trait analysis [28,29,30].

Paleontologists have also incorporated new knowledge from biology and genomics. As genomic sequencing became increasingly possible for a wide range of organisms, paleontologists began to combine morphological evidence from fossils with genomic data to reconstruct phylogenetic relationships [31,32,33].

Alongside the genomic revolution, there is another discipline in biology with significant implications for paleontology: elucidating the relationship between genotype and phenotype, often referred to as genotype:phenotype (G:P)-mapping. The insight that comes from G:P-mapping will fundamentally alter how we approach fossil morphologies in the 21st century and, consequently, improve our knowledge of the evolutionary past. To demonstrate this point, we investigated the insight that G:P-mapped dental traits bring to the African fossil record of monkeys (Primates: Cercopithecidae). We first used quantitative genetic analyses to assess the heritability and covariate effects on traditional measurements of tooth size and two types of G:P-mapped traits, one derived from developmental biology and the other from quantitative genetic analyses. We then compared how these traits vary across extant cercopithecids to test Hypothesis 1: G:P-mapped dental traits can provide evidence of phylogenetic history and selection, and therefore, are useful in paleontological investigations. We then focused on the traits defined through our quantitative genetic approach and explored how they vary in the fossil record to test Hypothesis 2: G:P-mapped traits reveal a range of morphological variation that cannot be predicted solely through extant variation.

## 2. Background: Traditional and G:P-Mapped Dental Traits

Paleontologists have long relied on the size of the postcanine teeth (especially the molars) to serve as a proxy for body size, to provide essential insight into taxonomy, and to observe patterns of evolution [34,35,36,37]. Tooth size is traditionally defined as the two-dimensional occlusal area of the crown, calculated by multiplying the mesiodistal length by the buccolingual breadth (Figure 1C). This trait has long been, and still is, an essential trait in mammalian paleontology.

Over the last couple of decades, technological advances in the biological sciences have enabled scientists to probe the genetic influences on tooth size variation. There are two main avenues for G:P-mapping of dental variation: quantitative genetics and developmental biology. Quantitative genetic analyses approach the G:P-map through phenotypic variation, investigating how anatomical variation is inherited through family lineages. So long as the family structure within a population is known, any taxon can be studied, including large-bodied and long-lived animals such as primates. Because quantitative genetics reveals the genetic contributions to phenotypic variation within a population, this approach is particularly informative for Neogene paleontology, as population-level variation is most applicable to micro-evolutionary questions [38,39]. In contrast, developmental approaches involve the manipulation of embryogenesis and organogenesis to gain insight into the formation of the dentition from a fertilized egg. Consequently, experimental developmental biology is limited to animals that are amenable to being raised in a laboratory setting, who have short generation times, and/or for whom organs can be grown in culture, such as mice.

While there is a deep history of quantitative genetic research on the dentition [40], results from recent analyses have clarified that individual teeth are not genetically or developmentally independent structures, and that different aspects of a tooth are underlain by different genetic and non-genetic influences. For example, minor shape variants on the crown are genetically independent of tooth size [41]. Looking along the dental arcade, we see that the size of the incisors is genetically independent from the size of the premolars and molars (in baboons [42]; and macaques [43]; with some suggestive evidence in humans [44,45]; but see tamarins [46,47], and a different study on humans [48]), yet there is significant pleiotropy between postcanine teeth [42,43,46,47,48]. Evidence of pleiotropy indicates a genetic correlation, meaning that a significant proportion of the residual phenotypic variance in the two traits is due to the shared additive effects of the same gene or set of genes. Thus, evidence of pleiotropy helps elucidate the underlying genetic architecture. Shared genetic effects are not just limited to within the dentition. In baboons, for example, we also discovered that molar width is genetically correlated with body size (with more than 20% of the additive genetic covariance between these traits estimated to be due to the same gene or set of genes), but in surprising contrast, molar length is not [49]. While this exact correlation has not yet been explored in other primates, variation in crown area for humans has a positive correlation with the length of the dental arch, and a negative correlation with arch width, suggesting that tooth area and size dimensions within human dentitions are similarly not uniform [48]. Based on this genetic evidence, we now know that variation in the 2D occlusal area (as studied by paleontologists) reflects a range of underlying genetic effects related to body size and sex in addition to the genetic effects that pattern dental variation.

In order to make this quantitative genetic evidence translatable to paleontological research, Hlusko and colleagues [38] developed two dental traits that reflect the genetic architecture of the baboon dentition: the molar module component (MMC) and the premolar-molar module (PMM). Both traits are based on our quantitative genetic analyses of baboon mandibular dental variation. These analyses revealed that the mesiodistal lengths of the first, second, and third molars share a genetic correlation that is essentially 100%, indicating that first, second, and third molars are, genetically speaking, not the separate, independent structures that anatomists have long viewed them to be, but rather, one organ [42,50,51]. Consequently, the relative mesiodistal lengths of the first, second, and third molars represent components within one genetic module. As mentioned previously, molar buccolingual width has significant pleiotropic effects on body size [49]. Therefore, Hlusko et al. [38] proposed the ratio of the mesiodistal length of the third molar divided by the mesiodistal length of the first molar as a trait (MMC) that captures the genetic variation influencing tooth size variation within the molar module without the genetic effects that also influence body size (Figure 1B). Consequently, MMC is a more direct reflection of the underlying genetic architecture influencing molar size variation than two-dimensional crown area (length × width) because 2-dimensional crown area results from a combination of genetic effects that include those that influence body size.

We also defined PMM as a ratio that reflects the genetic correlation between the size of the fourth premolar relative to the size of the molar module [38]. Previous analyses demonstrated that the mesiodistal length of the fourth premolar has an overlapping, but not complete genetic correlation with the mesiodistal length of the molars [42,50,51]. PMM is the mesiodistal length of the second molar divided by the mesiodistal length of the fourth premolar (Figure 1A). As with MMC, we focused on the mesiodistal lengths in order to avoid conflating the genetic effects on body size with those that influence dental patterning.

The mandibular versions of MMC and PMM were first identified for cercopithecid monkeys and then expanded to apes, revealing an episode of selection during the Late Miocene [38]. While we do not yet know the genetic mechanisms that underlie PMM and MMC, we do know that these two ratios reflect a genetic architecture that does not simultaneously influence body size or sex, and that appears to primarily influence variation in the relative sizes of teeth in the postcanine dentition of catarrhine primates [38,52] and many other mammals [53,54].

The influence of developmental mechanisms on two-dimensional molar size variation has also been explored. Kavanagh and colleagues [55] reported evidence of an inhibitory cascade within the molar teeth of mice that can explain variation in the relative sizes of the first, second, and third molars. Through experimental manipulation of cultured tooth germs, they found that the timing of first molar initiation influences the initiation time and ultimate size of the second and third molars. For example, the removal of the first molar bud led to earlier initiation of the second and third molars, and these later-forming teeth grew larger. Kavanagh and colleagues [55] observed that across murine rodents, the size of the second molar always accounts for approximately one-third of the two-dimensional size of the molar row in occlusal view, and that the relative sizes of the first and third molar vary around this. From these observations, they [55] proposed that evolution follows this rule of one-third, and that first and third molar size can be predicted from each other. This model is referred to as the inhibitory cascade (IC) model. The model fits well with the phenotypic variation observed across murines [55] and has been supported in a range of other mammals (e.g., early mammaliaforms [56]; kangaroos [57]; many but not all South American ungulates [58]; and many but not all rodents [59]). However, the IC model does not fit the patterns of variation observed for anthropoid primates [60,61], humans [62], and some earlier hominids [63].

For Hypothesis 1, we explore both types of G:P-mapped traits in the maxillary dentitions, the IC (from developmental biology), and the MMC and PMM (from quantitative genetics). For Hypothesis 2, we focus on the quantitative genetics-derived traits, complementing the previously published investigation of the mandibular versions of PMM and MMC with the maxillary analyses.

## 3. Materials and Methods

Our analyses rely on dental linear metrics from three different samples described in detail in the following paragraphs. The quantitative genetic analyses were performed on data from 611 individuals within a captive pedigreed population of *Papio hamadryas* baboons. The extant, neontological analyses were performed using data from 825 museum skeletal specimens representing 13 genera within Cercopithecidae. Finally, we augmented the data we collected from museum specimens with data culled from the published scientific literature to create a fossil dataset of 1,436 individuals from 17 genera representing the last 20 million years of cercopithecid evolution in Africa.

**Sample 1, quantitative genetics:** The baboons from which dental data used in our quantitative genetic analyses were obtained are members of a large, six-generation pedigree (*n* = 2426), developed and maintained at the Southwest National Primate Research Center (SNPRC) at the Texas Biomedical Research Institute (Texas Biomed) in San Antonio, Texas. The pedigree was genetically managed to minimize inbreeding, and ascertainment of animals for this study was random with respect to phenotype. We analyzed linear crown metric data for the maxillary fourth premolar and first, second, and third molars obtained from 611 members of the single, large, six-generation pedigree. The female to male sex ratio was approximately 2:1 and the mean age of the sample was approximately 16 years, with ages ranging from 8 to 32 years. All procedures involving animals were reviewed and approved by Texas Biomed’s Institutional Animal Care and Use Committee. SNPRC facilities and animal use programs at Texas Biomed are accredited by the Association for Assessment and Accreditation of Laboratory Animal Care International, comply with all National Institutes of Health and U.S. Department of Agriculture guidelines, and are directed by Doctors of Veterinary Medicine.

**Sample 2, extant variation:** Our comparative sample of extant taxa includes 825 individuals (Table 1). Most of the extant comparative data were collected by the authors and have been included in previously published research [64]. This dataset builds on the published dataset [65].

**Sample 3, extinct variation:** Our comparative sample of fossil taxa includes 1436 individuals (Table 2). Fossil data include measurements collected by the authors, culled from published sources, and downloaded from PRImate Morphometrics Online (PRIMO). Data sources for each sample are specified in Table 2.

**Data collection:** Tooth dimensions for the SNPRC baboons are described in Hlusko et al. [76]. For the other two samples, mesiodistal length and buccolingual breadth measurements were collected from the maxillary fourth premolar (P4) and the three maxillary molars (M1, M2, and M3) for each individual, for both left and right sides, following standard protocols (see [64]). For the measurements collected by our research team, we did not account for interstitial wear. For the data culled from other publications, we refer to those publications, noting that some authors do not explicitly state how they measured mesiodistal length on teeth with significant interstitial wear. We used these two linear measurements, mesiodistal length (L) and buccolingual breadth (W) (see inset of Figure 1), to calculate 2-dimensional occlusal area, MMC, PMM, and the IC (see Figure 1 for equations).

**Abbreviations:** Premolars are abbreviated as P, molars as M. The letter for the tooth (P or M) is followed by a number indicating tooth position. For example, M2 refers to the second molar. We are primarily focused on a discussion of maxillary molars in this manuscript. We specifically indicate if a measurement or tooth is from the mandibular dental arch in the text rather than through abbreviations.

**Overview:** In order to test Hypothesis 1, we first established that a significant proportion of the phenotypic variation in all of the six traits is attributable to the effects of genes, i.e., that all the traits are heritable. To do this, we estimated the heritability of the traits in the SNPRC baboons. We then assessed the variation of all six traits across a sample of extant cercopithecid monkeys and considered how they vary within a phylogenetic context through a phylogenetic ANOVA. We followed the ANOVA with an analysis to test whether the traits are phylogenetically conserved or show evidence of selection. For the test of Hypothesis 2, we focused on the two traits derived from quantitative genetics: PMM and MMC. We first reconstructed ancestral states (ASR) based on the phylogenetic relationships of the extant genera analyzed for Hypothesis 1. We then compared the ASR trait values derived from the extant taxa to the PMM and MMC values observed in the fossil record.

**Quantitative genetic analyses:** We conducted statistical genetic analyses using a maximum likelihood-based variance decomposition approach implemented in the computer package SOLAR ([77]; v 8.1.1, www.solar-eclipse-genetics.org). This approach partitions the observed covariance between individuals into genetic and environmental components. The variance components are additive, with the phenotypic variance ($\sigma_{P}^{2}$) being the sum of the genetic ($\sigma_{G}^{2}$) and environment ($\sigma_{E}^{2}$) variances. Estimates of heritability (*h*^2^), the proportion of the phenotypic variance attributable to additive genetic effects, were obtained as: $$h^{2}=\sigma_{G}^{2}/\sigma_{P}^{2}$$

Unless otherwise noted, all quantitative genetic analyses were conducted following inverse gaussian normalization of the residuals (trait values were adjusted for the mean effects of sex and/or age, the latter a rough proxy for wear, if significant). Significance of the maximum-likelihood estimates for heritability and other parameters was assessed by means of likelihood ratio tests [78]. The maximum likelihood for a general model in which all parameters were estimated was compared to that for restricted models in which the value of the parameter to be tested was held constant (value dependent on null hypothesis). Twice the difference in the log-likelihoods of the two models compared is distributed asymptotically approximately as either a 1/2:1/2 mixture of χ^2^ with a point mass at zero for tests of parameters such as h^2^ for which a fixed value of zero in a restricted model is at a boundary of the parameter space or a χ^2^ variate for tests of covariates for which zero is not a boundary value [79]. In both cases, degrees of freedom are obtained as the difference in the number of estimated parameters in the two models [79]. However, in tests of parameters such as h^2^, where values may be fixed at a boundary of their parameter space in the null model, the appropriate significance level is obtained by halving the *p*-value [80]. 

**Descriptive statistics:** Statistical analyses were completed in the R statistical environment v3.2.2 [81]. We first calculated univariate descriptive statistics for the two-dimensional areas, IC, MMC, and PMM values for all taxa included in the study, using built-in functions in R. Kurtosis was calculated using the *moments* package in R [82]. We visualized the distribution of the MMC and PMM traits across taxa in R using the package *ggplot2* (v1.0.1; [83]).

**Phylogenetic ANOVA:** We conducted a phylogenetic ANOVA to investigate variation across cercopithecid genera using the aov.phylo function in *geiger* [84]. The phylogenetic ANOVA uses average species data to compare traits across genera. Analyses were run on left side maxillary data. When no left side data were available, the right side was included. All dental areas were geometric mean size-corrected prior to analysis. All other dental traits are unit-free ratios.

**Phylogenetic analyses:** For all phylogenetic analyses, we used a consensus molecular chronogram based on a Bayesian phylogenetic analysis of genetic data downloaded from the 10kTrees v.3 database, built using data from six autosomal genes and 11 mitochondrial genes sampled from GenBank [85]. *Presbytis rubicunda* is not available in the 10kTrees database, and so we added this taxon manually to the phylogeny in R using a branch length split age of 1.3 million years from *Presbytis melalophos* [38,86].

**Test of phylogenetic signal and selection:** We tested the phylogenetic signal of the dental traits with a Blomberg’s K analysis using phylosignal in *picante* [87]. Blomberg’s K tests whether a trait is present in closely related taxa more frequently than would be expected by Brownian motion [88]. The K value for a trait can be either less than 1, equal to 1, or greater than 1. A K value > 1 is generally interpreted as more phylogenetically conserved than expected under neutral Brownian motion, while a K value of 1 generally indicates Brownian evolution of the trait under drift. In contrast, K < 1 is generally interpreted as a trait that is phylogenetically conserved, although less so than expected under a Brownian model, suggesting that selection pressures may be influencing the distribution of the trait in ways that deviate from the pattern expected based on phylogeny (with K = 0 implying that a trait varies in a pattern completely unrelated to phylogeny). However, heterogeneous rates of genetic drift or rapid divergence between species can also result in low K values [88,89]. We used summary trait values for each species and compared average species values across genera.

**Ancestral state reconstruction:** To investigate how dental traits have evolved in cercopithecids, we generated a series of ancestral state reconstructions (ASR) using contMap in *phytools* [90], which maps continuous variables across a phylogeny. We quantified the estimated values at internal nodes using fastAnc in *phytools* [90], a function that generates maximum likelihood ancestral states for continuous traits.

## 4. Results

### 4.1. Test of Hypothesis 1: G:P-Mapped Dental Traits Can Provide Evidence of Phylogeny and Selection

The results of the quantitative genetic analyses are presented in Table 3. Statistically significant residual h^2^ estimates, ranging from 0.611 to 0.728, were obtained for five of six two-dimensional areas, two on the left side and three on the right. Both sex and age exerted significant mean effects on the two left side 2-dimensional areas, while only sex influenced the three right side traits. These covariate effects were substantive, accounting for approximately 28% to 51% of the total phenotypic variance in these five 2-dimensional areas. These same analyses returned significant h^2^ estimates (range: 0.491–0.604) for three of the six G:P-mapped traits: right IC, and right and left PMM, with sex being the lone significant covariate, accounting for approximately 2% to 9% of their total phenotypic variance.

The analyses did not return statistically significant heritability estimates for four phenotypes, three on the left side of the arch (M3 2D area, IC, MMC) and one on the right (MMC). Derivation of these traits was based on data from comparatively small numbers of animals: i.e., only 140 to 221 individuals of the more than 600 pedigreed baboons from which data were obtained for this study.

**Extant variation descriptive statistics:** Univariate statistics for the two-dimensional areas of M1, M2, and M3, and the G:P-mapped traits (IC, PMM, and MMC) are reported in Table 4 and Table 5. These are based on the phenotypic observations of the taxa listed in Table 1. See Appendix A for more detailed descriptive statistics (Appendix A).

**Phylogenetic ANOVA:** Results from the phylogenetic ANOVA are presented in Table 6. The summary *p*-values indicate that all six traits differ significantly across the genera included in the analyses. The *p*-values for each genus are also presented. For two-dimensional areas, *Nasalis*, *Colobus*, *Macaca*, *Lophocebus*, and *Erythrocebus* are not different from the pooled value of the trait across all the extant genera. *Piliocolobus* is only statistically different for the M2. *Chlorocebus* is only statistically different for the M2 and M3 two-dimensional areas. IC and MMC results are identical, demonstrating that *Cercopithecus*, *Mandrillus*, *Papio*, and *Theropithecus* are statistically significantly different from the pooled values of IC and MMC. PMM differentiates most of the papionins (*Macaca*, *Papio*, and *Theropithecus*) as well as the colobine *Nasalis* from the other genera.

**Phylogenetic signal:** Blomberg’s K-values for the six traits are reported in Table 7. These all range between 0.625 and 0.673. Statistically non-significant *p*-values indicate that the trait is evolving neutrally under Brownian motion. IC is marginally significant at the *p* = 0.05 level, and therefore may indicate that IC variation observed across these extant taxa is the result of selection. MMC is statistically significant at the *p* = 0.05 level, providing a clear indication that selection has likely been operating on the relative mesiodistal lengths of the molars. Blomberg’s K is a conservative test that is sensitive to sample size [88]. Additionally, variation in sample sizes across taxa, as well as variation in sample source populations within taxa, have been demonstrated to skew mean trait values used in these analyses, which can in turn skew results [91]. Sampling more extensively within sparsely sampled taxa, and across a broader range of primate taxa, may reveal stronger phylogenetic signal for these traits.

### 4.2. Test of Hypothesis 2: G:P-Mapped Traits Reveal a Range of Morphological Variation That Cannot Be Predicted Solely through Extant Variation

**Ancestral State Reconstruction (ASR):** ASR estimates based on the extant genera listed in Table 1 are presented in Table 8, with nodes defined on the molecular phylogeny shown in Figure 2.

**Comparison to fossil data:** In order to compare the ASR trait values to the anatomical variation observed in the fossil record, we compiled data for 17 fossil genera (Table 2) that could possibly be a fossil representative for one of the ASR nodes (Table 8). We include the molecular divergence date estimates that correspond to each node in the phylogeny. Next to these data, we list the possible fossil representative genus, along with the MMC and PMM values associated with that genus and the associated geological age range. Note that some fossil genera are potentially associated with more than one node. We present these data visually in Figure 3, along with the extant data for comparison. The averages for the fossil genera are indicated with a skull icon. Each fossil data point is linked with a double-ended arrow to the ASR node/estimate it may potentially represent, highlighting the difference between them. For both the PMM and MMC, the ASR estimates are usually lower than the values observed in the fossils. We present the absolute value of the difference between the ASR trait estimate and the fossil trait in Figure 4. Absolute value of the average difference between ASR MMC and fossil MMC is 0.066. Absolute value of the average difference between ASR PMM and fossil PMM is 0.162. At all of the time points represented by these data, the difference between the ASR value and the fossil value is most distinct for PMM.

## 5. Discussion

As advances in genetics and developmental biology make it possible to elucidate the relationship between genotype and phenotype (G:P), paleontologists are able to modify their approaches to anatomical variation accordingly. Our aim in this study was to understand how the method of trait definition influences the ability to reconstruct phylogenetic relationships and evolutionary history inCercopithecidae, the Linnaean Family of monkeys currently living in Africa and Asia. We compared one of the most classic traits in primate paleontology, two-dimensional occlusal tooth size (calculated as the mesiodistal length of the crown multiplied by the buccolingual breadth), to a trait that reflects developmental influences on molar development (the inhibitory cascade, IC [55]) and two traits that reflect the genetic architecture of postcanine tooth size variation defined through quantitative genetic analyses: MMC and PMM [38].

We first established that our maxillary trait types are highly heritable (albeit sensitive to low sample sizes), indicating that variation in tooth size, however it is assessed, is significantly influenced by genetic variation. This result was expected, as it builds on many decades of quantitative genetic analyses of dental variation demonstrating that tooth size is one of the most heritable phenotypes (e.g., [40]). At first glance, there are two caveats to this conclusion. First, while the right IC heritability estimate is significant, the left is not. We know from past analyses that antimeres (left and right side corresponding traits) generally return genetic correlations of one, indicating that they are influenced by identical genetic effects [41,42,50,51,100]. Therefore, we are confident that the left IC is also heritable, similarly to the right, and that our analysis is just underpowered by the small sample size. The second caveat is that we found that both left and ride side maxillary MMC traits returned non-significant heritability estimates. This was not unexpected given the small number of individuals (*n* = 191 for the left and 140 for the right) with data available. We are confident that this non-significant result is due to the analysis being underpowered rather than a true biological signal, given that the component dimensions when analyzed individually are highly heritable [42,50,51], and that the mandibular homologue of this trait is significantly heritable [38]. However, that said, further analyses with larger sample sizes are clearly needed.

These quantitative genetic analyses provide a good example of how challenging this approach can be, and why this type of research within evolutionary biology is only now becoming more common. Sampling is a significant challenge. For example, in our data set for the SNPRC baboons, composite traits reduce the number of individuals that can be included by a remarkable degree, especially for traits that include measurements of the third molar. We see this data reduction because the SNPRC measurements were collected from dental casts made of living animals. Consequently, the gumline often obscures the back edges of the third molar. Therefore, in a sample of 611 animals within the SNPRC colony, we only have M3 mesiodistal lengths for 140 (right side) and 191 (left side) individuals. Another significant factor in the success of quantitative genetic analyses is the location of the individuals within the pedigree. For example, even though we have more SNPRC baboon individuals available for the analysis of the left IC (*n* = 170) compared to the right (*n* = 127), only the right value returned a significant heritability estimate for IC. This is likely the result of where those individuals with data fall in the pedigree rather than evidence of a different biological signal. We are currently in the process of expanding the SNPRC dental data set and anticipate revisiting these analyses with a larger sample size.

Ever since Darwin [101], biologists have recognized that the heritable nature of phenotypic variation is central to the theory of evolution by natural selection. While all paleontologists appreciate this fact, ascertaining heritability is not simple. Even though the fundamental concept of quantitative genetics originated with Mendel, the ability to analyze the inheritance of normal, continuously varying traits across complex pedigrees was not possible until recently, as the algorithms are computationally intense and require modern computing technologies (for a history of approaches to dental variation: [40]). The modern concepts of evolutionary quantitative genetics were developed almost forty years ago [102,103,104,105], but it has been over the last 20 years that there has been an incredible expansion of quantitative genetic analyses being applied to evolutionary questions (examples of this research using primate models: [38,43,46,47,48,49,100,106,107,108,109,110]).

In addition to the high heritability estimates, we also find that G:P-mapped traits are phylogenetically conserved and show evidence of selection. ANOVA indicates that all six traits vary significantly across the cercopithecid clade, however, there are interesting differences in how variation in these traits is distributed across the Linnaean families, tribes, and genera. Within the colobines, *Presbytis* is significantly different in terms of two-dimensional molar size from other colobines, but not for the G:P-mapped traits. Previous researchers noted that the maxillary M3 morphology and eruption sequence of *Presbytis* sets it apart from other Asian colobines [111,112]. The lack of significant variation in the G:P-mapped traits for *Presbytis* poses the hypothesis that the distinct M3 morphology of *Presbytis* compared to other Asian colobines is not due to variation in the dental genetic architecture of PMM, MMC, or IC. Perhaps the unusual *Presbytis* dental morphology is related to body size, as the two-dimensional areas that are significantly different have pleiotropic effects with body size variation, possibly related to degrees of evolutionary dwarfism in this genus [64,113].

The ANOVA also revealed a distinct separation of three of the papionin genera: *Papio*, *Theropithecus*, and *Mandrillus*. These three genera are derived among the cercopithecids in having elongated muzzles, which is well-known to demonstrate positive allometry [114,115,116,117]. Looking more closely, we see that *Papio* and *Theropithecus* differ from the other genera in all six dental traits. However, *Mandrillus* differs in the two-dimensional area traits and the IC and MMC, but not PMM. Given that *Mandrillus* may be in a clade more closely related to *Macaca* than *Papio*/*Theropithecus*/*Lophocebus* [93,118], our results suggest that the phenotypic expression of MMC and IC are convergent in these two clades, and that the expressions of PMM differ despite the similarity in overall muzzle elongation. Previous in-depth analysis of the morphological variation of the faces of *Mandrillus* and *Papio* supports the interpretation that their elongated muzzles are convergent [115]. Our G:P analysis offers the first glimpse into the possible genetic mechanisms that may have been co-opted in this example of parallel evolution.

As described in the Introduction, the MMC and the IC are similar conceptually but distinct in their implementation and aims. The “inhibitory cascade” is a model proposed to explain the pattern of molar size variation observed across murines [55]. The IC model is based on the observation that the timing of initiation of the posterior molars is modulated by the growth of the first molar [55], confirming previous research. Lumsden and Osborn [119] and Lumsden [120] observed that all three molars develop from the ectopic transplantation of just the mouse M1 germ. By measuring the daily growth of mouse molars from 14 to 23 days post-fertilization, Sofaer [121] found compensatory changes in growth rate that seem to result from “some kind of competitive interaction” between the molars [121]. Lucas et al. [122] also observed that for 67 primate species, the size of the maxillary M2 is stable in accounting for 33–40% of the size of the molar row, with the M1 and M3 varying around the M2 in a compensatory manner. Kavanagh et al. [55] provided more experimental evidence for the mechanism first identified by the earlier investigators, gave it a name, and tested the model across the dental variation within Murinae. Since then, the authors have extended it to be a “simple rule govern[ing] the evolution and development of hominin tooth size” [61,123].

When the MMC and PMM were first proposed, we described the variation captured by MMC as likely due to the same developmental mechanisms underlying the IC [38]. However, we named our measurement in terms of the anatomical structures being assessed (the components of the molar genetic module) rather than by a hypothetical developmental mechanism [55], the genetics of which have not yet been established to our knowledge. Therefore, the MMC is not a developmental model likethe IC (contra [124]), but rather a measurement protocol for assessing molar size variation. As Lucas et al. [122] noted, the M1/M3 ratio is a measure of the shape of the tooth row. IC and MMC both capture this shape variation through ratios, but with a distinct difference. The IC is based on the two-dimensional size of M3 divided by the two-dimensional size of M1 (the traditional anatomical assessment of tooth size). In contrast, the MMC is the ratio of the length of M3 divided by the length of M1, which focuses the ratio on the genetic effects that result in variation in the relative lengths of the molars, separatefrom the genetic effects that influence molar width and also body size [38,49]. This distinction between the genetic architecture of length and width dimensions accords with Sofaer et al. [125]’s conclusion that mesiodistal lengths and buccolingual widths are influenced by different genetic and environmental effects, as well as Marshall and Corrucini [126]’s observation that molar lengths change much more slowly than widths in marsupial lineages with evolutionary dwarfing. Based on all of this evidence, the MMC is in all likelihood a more precise reflection of the genetic patterning mechanism that influences molar size proportions in cercopithecids, if not primates and other mammals more generally, compared to the IC.

Our analyses presented here further support the interpretation that the IC and MMC overlap in the genetic influences on molar size variation that they capture. For example, in the quantitative genetic analyses, the IC, MMC, and PMM all have much smaller covariate effects compared to two-dimensional areas (0.05 on average compared to 0.38 on average, respectively). Additionally, the IC and MMC have the same pattern of significance across genera in our ANOVA. This molar module pattern is distinct from the PMM, providing additional evidence that PMM is capturing a genetic mechanism distinct from that of the MMC (and IC). Our estimation of Blomberg’s K also reveals similarities between MMC and IC. However, the results presented here suggest that our measurement protocol for MMC may well be a more specific reflection of the underlying genetic mechanism influencing molar proportions in cercopithecids compared to IC, given that we removed the known pleiotropic effects with body size. Further genetic analyses are needed to explore this with more certainty.

There has been a lot of enthusiasm for what G:P-mapped dental traits might offer for oral health [127] as well as paleontology (e.g., [128,129]). Evans (of [123]) even suggested that for hominids “This pattern is so strong, we can predict the size of the remaining four teeth without even finding the fossils!” (http://evomorph.org/inhibitory-cascade, accessed on 17 July 2022). With evolutionary biologists expressing this type of sentiment about the utility of fossils, it would not be unreasonable for funding agencies and budding scientists to ask if field paleontology is a thing of the past. Does the future of paleontology need new fossils?

In light of this question, our second major aim was to investigate G:P-mapped traits within the fossil record. For this, we focused on the maxillary MMC and PMM and compared computer-generated estimates of ancestral traits to the traits observed on fossils. We want to be up front about there being no clear consensus on direct ancestor-descendant relationships among cercopithecids over the last five million years, as the African cercopithecids from the Plio-Pleistocene are remarkably different from extant monkeys [95,99]. Consequently, new approaches are clearly needed, and G:P-mapped traits might offer novel insight into this murky evolutionary history.

Our comparisons of the ASR estimates with the fossil values unequivocally demonstrate that ASR based on extant data is compromised by the phenomenon of “the tyranny of the present”. The lure of the extant comparative data available in museum collections unintentionally limits our expectations for what ancestral morphologies could have been. For example, we find that both MMC and PMM ASR estimates return values lower than what is observed in the fossil record penecontemporaneous with the ancestral nodes (Figure 3). PMM is underestimated twice as much as is MMC (Figure 4). Anecdotally, Figure 3 shows that ASR essentially averages the observed variation and is therefore unable to predict a wider range of variation than that of the input. While paleontologists are sometimes able to input fossil morphologies into their analyses to avoid this bias (e.g., [130]), this requires a high degree of confidence in the ancestor-descendant relationships, something we do not have for the Cercopithecidae. For monkeys, the modern bias in ASR would lead to the interpretation of the PMM of *Papio* and *Theropithecus* as newly derived, when we see that they actually have quite similar PMM values to early papionin genera such as *Parapapio*, *Pliopapio* and *Soromandrillus*. The high MMC values of the Miocene and Pliocene colobines also change how we view the evolutionary relationship of the African and Asian colobines. Knowing that earlier colobines in Africa had higher MMC values than both extant African and Asian colobines suggests that the African and Asian colobines evolved along the same MMC trajectory (reducing the MMC over time). None of these trends are visible when just size alone is considered.

The next step is to figure out what genetic mechanisms MMC (and IC) and PMM capture. We have a few hints. Previous analyses have shown that mandibular MMC is likely more evolutionarily conserved than PMM within catarrhine primates [38], across Boreoeutheria [53], between the different genera of megabats [54], and in the fossil record of the hominids [52]. Our results here for cercopithecids similarly demonstrate that the genetic mechanism captured by maxillary PMM appears to be more evolutionarily labile than maxillary MMC. We report elsewhere that variation in MMC may covary with prenatal growth rates [131], and therefore, MMC, a dental trait, may actually reflect life history variation rather than mastication and diet. If future analyses bolster this conclusion, G:P-mapping of dental variation opens a new window to the paleobiologies preserved in fossil morphology. But without the fossil evidence, we will never fully understand the range of variation that has existed over the evolutionary history of the Cercopithecidae. Therefore, the discovery of new fossils is not only still relevant, but even more revelatory as we apply 21st century methods to this most ancient data set.

## Figures and Tables

**Figure 1 biology-11-01218-f001:**
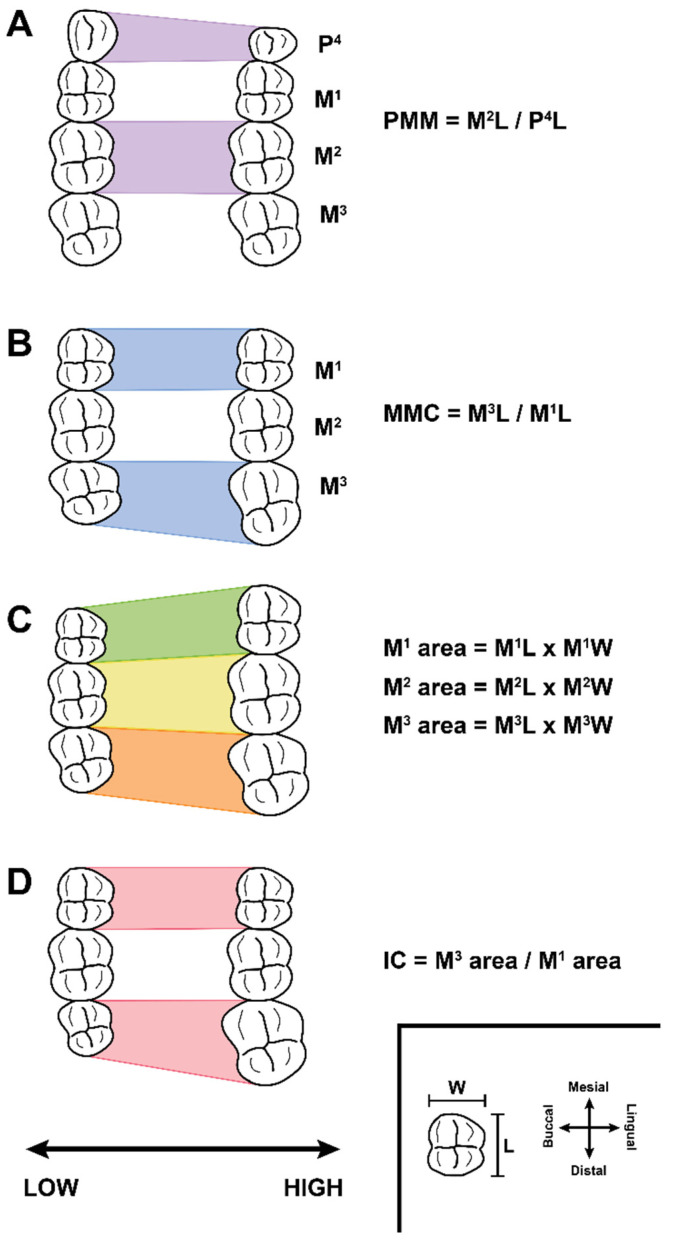
Illustration of the four maxillary dental traits investigated in this analysis. Each panel shows the right maxillary occlusal view of two extremes for one of the traits. Mesial is to the top, distal to the bottom, lingual to the right, and buccal to the left. The axis at the bottom of the figure orients the reader to how the morphology varies according to low and high values of the trait. (Panels (**A**,**B**)) demonstrate the two traits defined through quantitative genetic analyses, ratios that reflect the relative size variation between the premolar and molar genetic modules (PMM; panel (**A**)) and the relative sizes of the molars within the molar module (MMC; panel (**B**)). (Panel (**C**)) shows the traditional method for studying molar size variation within paleontology, by calculating a two-dimensional area of the occlusal view of the crown. (Panel (**D**)) shows the “inhibitory cascade” (IC) trait, defined through developmental gene expression studies of mice. See text for more detailed descriptions.

**Figure 2 biology-11-01218-f002:**
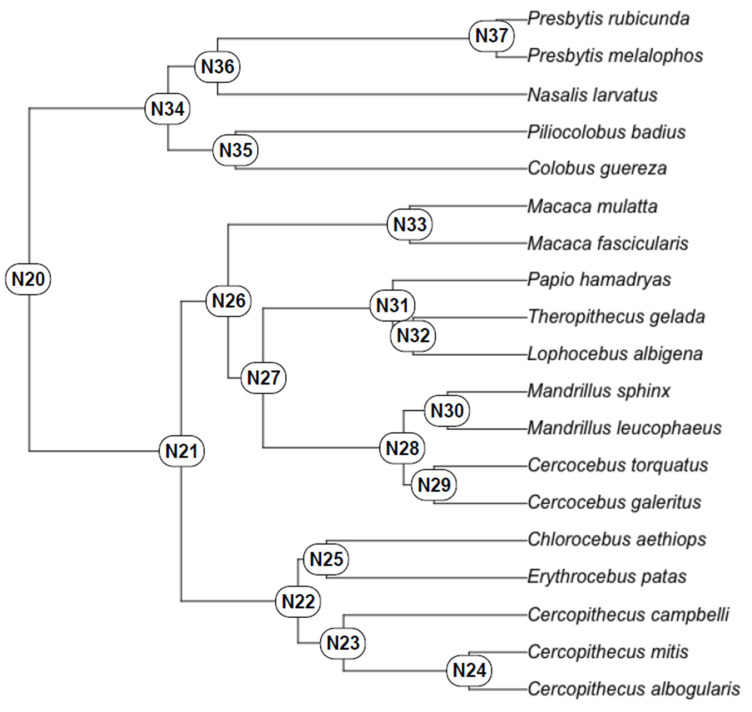
Molecular phylogeny of the extant cercopithecid genera included in this analysis with ASR nodes indicated. See Table 8 for ASR MMC and ASR PMM estimates.

**Figure 3 biology-11-01218-f003:**
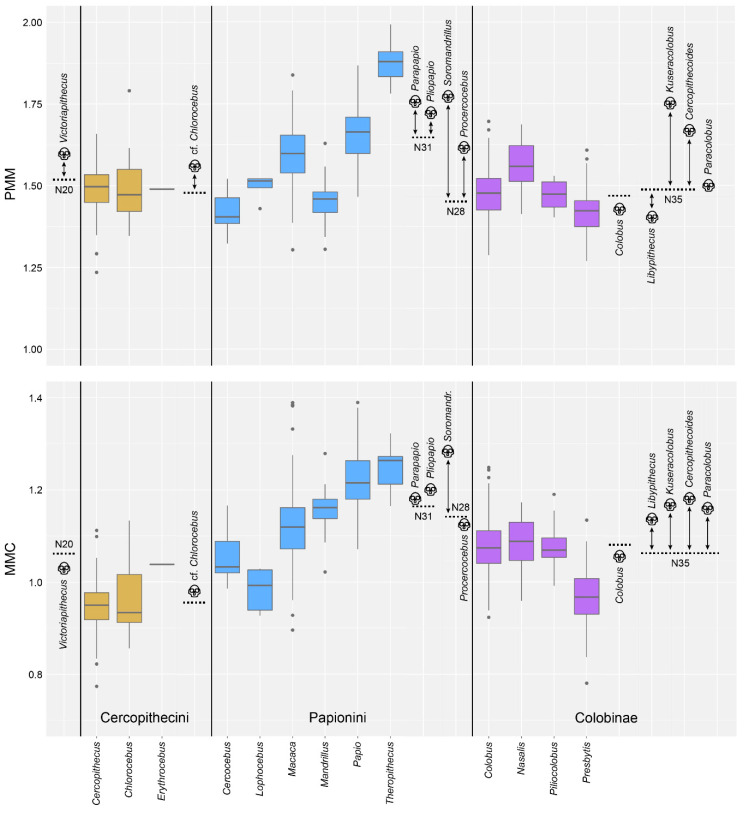
Box and whisker plots showing the range of variation for PMM and MMC within the sampled extant genera (labeled at the bottom of the figure). The genera are color-coded, with tribe Cercopithecini in gold, tribe Papionini in blue, and the subfamily Colobinae in purple. In addition to the extant data, we plot trait estimates for the Ancestral State Reconstruction (ASR) nodes as horizontal dotted lines, labeled with N and the number of the node. The possible fossil representatives for these nodes are plotted within the tribe or subfamily to which the fossil belongs. *Victoriapithecus*, on the far left, is widely thought to be ancestral to the split between the Colobinae and the Cercopithecinae (which includes Cercopoithecini and Papionini, shown here) [99]. Notice that for all but two of the PMM ASR-fossil pairs, the ASR estimate is lower than the observed fossil values. Similarly, for all but two of the MMC ASR-fossil pairs, the ASR estimate is also lower than the observed values. These differences are shown quantitatively in Figure 4.

**Figure 4 biology-11-01218-f004:**
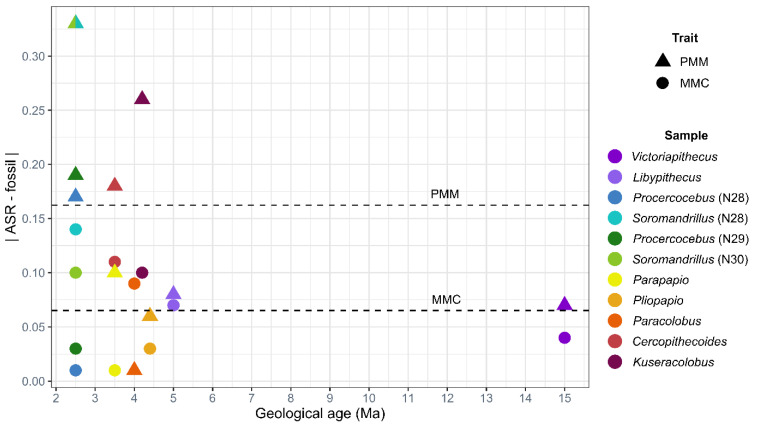
Bivariate plot of the difference between ASR trait values and fossil evidence for PMM and MMC. Geological age is shown on the *X*-axis. On the *Y*-axis, we report the absolute value of the difference between the ASR-estimated trait value for each node (molecular divergence) and the trait values observed for the African cercopithecid fossil genera in the same Tribe living near the time of the molecular divergence. The genera are shown in separate colors, defined in the key to the right. Triangles represent the PMM trait, and circles represent the MMC trait. The average difference for PMM is indicated by the top dashed line. The average difference for MMC is indicated by the lower dashed line. *Procercocebus* and *Soromandrillus* are included twice, as they could represent the ancestral morphology for nodes 28, 29, and 30.

**Table 1 biology-11-01218-t001:** Taxonomic composition of the extant comparative dataset.

Subfamily	Tribe	Genus	Species	Number of Individuals
Cercopithecinae	Cercopithecini	*Cercopithecus*	*albogularis*	1
		*Cercopithecus*	*campbelli*	9
		*Cercopithecus*	*mitis*	95
		*Chlorocebus*	*aethiops*	28
		*Erythrocebus*	*patas*	2
	Papionini	*Cercocebus*	*atys*	4
		*Cercocebus*	*galeritus*	1
		*Cercocebus*	*torquatus*	20
		*Lophocebus*	*albigena*	3
		*Macaca*	*fascicularis*	98
		*Macaca*	*mulatta*	76
		*Mandrillus*	*leucophaeus*	1
		*Mandrillus*	*sphinx*	17
		*Papio*	*hamadryas*	127
		*Theropithecus*	*gelada*	10
Colobinae	Colobini	*Colobus*	*guereza*	125
		*Nasalis*	*larvatus*	30
		*Piliocolobus*	*badius*	15
	Presbytini	*Presbytis*	*melalophos*	83
		*Presbytis*	*rubicunda*	80
TOTAL				825

**Table 2 biology-11-01218-t002:** Taxonomic composition of the fossil comparative dataset *.

Subfamily	Tribe	Genus	Species	Number of Individuals	Source
Cercopithecinae	Cercopithecini	*Cercopithecus*	sp. (Andalee)	30	1
		*Cercopithecus*	sp. (Upper Andalee)	5	1
		cf. *Chlorocebus*	Asbole	13	2
		cf. *Chlorocebus*	sp. (Chai Baro)	105	3
		cf. *Chlorocebus*	sp. (Faro Daba)	223	3
	Papionini	*Papio*	*hamadryas angusticeps*	12	4, 5
		*Papio*	*hamadryas robinsoni*	29	4, 6
		*Papio*	*hamadryas* ssp. (Asbole)	10	2
		*Papio*	*hamadryas* ssp. (Chai Baro)	143	3
		*Papio*	*hamadryas ursinus*	1	4
		*Papio*	*izodi*	7	4, 6, 7
		*Parapapio*	*broomi*	34	4, 6, 7
		*Parapapio*	*jonesi*	12	4, 7
		*Parapapio*	*whitei*	16	4, 6, 7
		*Pliopapio*	*alemui*	5	8
		*Procercocebus*	*antiquus*	8	6, 7
		*Soromandrillus*	*quadratirostris*	11	9
		*Theropithecus*	*oswaldi* cf. *darti*	124	10
		*Theropithecus*	*oswaldi darti*	4	7
		*Theropithecus*	*oswaldi leakeyi*	12	2, 11
		*Theropithecus*	*oswaldi oswaldi*	8	4
Colobinae	Colobini	*Cercopithecoides*	*kimeui*	12	9, 12
		*Cercopithecoides*	*meaveae*	2	9
		*Cercopithecoides*	*williamsi*	91	9
		*Colobus*	cf. *guereza* (Faro Daba)	360	3
		*Colobus*	sp. (Andalee)	31	1
		*Colobus*	sp. (Asbole)	47	2
		*Colobus*	sp. (Upper Andalee)	4	1
		*Kuseracolobus*	*aramisi*	5	8
		*Kuseracolobus*	*hafu*	14	13
		*Libypithecus*	*markgrafi*	3	9
		*Microcolobus*	*tugenensis*	1	9
		*Paracolobus*	*chemeroni*	1	9
		*Paracolobus*	*enkorikae*	8	14
		*Paracolobus*	*mutiwa*	22	9
		*Rhinocolobus*	*turkanaensis*	23	9
Victoriapithecinae		*Victoriapithecus*	*macinnesi*	40	15
			TOTAL	1436	

* Data sources: 1 [66]; 2 [67]; 3 (Authors measured at the National Museum of Ethiopia); 4 (Authors measured at the Ditsong Museum of Natural History); 5 [68]; 6 (Authors measured at the University of California Museum of Paleontology); 7 (Authors measured at University of the Witswatersrand); 8 [69]; 9 (PRIMO); 10 [70]; 11 [71]; 12 [72]; 13 [73]; 14 [74]; 15 [75].

**Table 3 biology-11-01218-t003:** Residual heritability estimates for the three types of maxillary dental traits: two-dimensional area, IC, MMC, and PMM *.

Trait	h^2^	h^2^ se	*p*-Value	Number of Individuals	Proportion of Variance Due to Covariates
LM1 2D area	0.611	0.110	<0.001	461	0.353 **
LM2 2D area	0.728	0.091	<0.001	537	0.413 **
LM3 2D area	0.261	0.190	0.260	221	0.490 *
RM1 2D area	0.703	0.132	<0.001	440	0.281 *
RM2 2D area	0.681	0.103	<0.001	531	0.366 *
RM3 2D area	0.726	0.275	0.004	171	0.508 *
L IC	0.181	0.156	0.100	170	0.103 *
R IC	0.604	0.333	0.006	127	0.094 *
L MMC	0.001	0.141	0.496	191	0.082 **
L PMM	0.491	0.093	<0.001	402	0.022 *
R MMC	0.238	0.228	0.096	140	0.044 *
R PMM	0.527	0.114	<0.001	380	0.030 *

* L = left; R = right; 2D = 2-dimensional; M1, 2 or 3 = first, second, or third molar; IC = inhibitory cascade trait; MMC = molar module component ratio; PMM = premolar-molar module ratio. * indicates sex only is a significant covariate. ** indicates sex and age are significant covariates. Shaded rows are statistically non-significant at *p* < 0.05.

**Table 4 biology-11-01218-t004:** Descriptive statistics for the two-dimensional area traits *.

	M1 Area		M2 Area		M3 Area	
	Mean (n)	StDv	Mean (n)	StDv	Mean (n)	StDv
**Extant Genera**						
** *Cercocebus* **	59.17 (15)	4.79	70.55 (15)	6.91	65.30 (15)	8.64
** *Cercopithecus* **	32.75 (91)	5.75	40.44 (96)	5.16	31.59 (88)	4.90
** *Chlorocebus* **	34.56 (27)	9.13	42.58 (27)	10.88	35.08 (25)	12.47
** *Colobus* **	45.31 (110)	5.11	55.19 (119)	5.97	51.88 (105)	6.26
** *Erythrocebus* **	39.29 (1)	-	48.16 (1)	-	47.06 (1)	-
** *Lophocebus* **	47.32 (6)	2.21	58.06 (6)	3.86	49.32 (5)	4.45
** *Macaca* **	44.26 (174)	8.30	58.91 (174)	12.73	55.94 (148)	12.40
** *Mandrillus* **	95.96 (18)	10.64	130.4 (18)	12.86	132.03 (18)	12.29
** *Nasalis* **	49.24 (30)	5.49	59.35 (30)	4.58	54.79 (30)	5.36
** *Papio* **	161.85 (84)	23.28	230.63 (106)	34.04	242.68 (104)	40.53
** *Piliocolobus* **	39.36 (15)	2.42	44.78 (15)	2.56	45.32 (15)	3.23
** *Presbytis* **	32.11 (151)	2.39	34.24 (153)	2.70	30.86 (145)	3.05
** *Theropithecus* **	96.15 (9)	11.11	140.87 (9)	11.61	145.15 (9)	11.66
**Fossil Genera**						
** *Cercopithecoides* **	78.92 (22)	16.74	104.41 (25)	22.32	98.85 (22)	19.59
** *Cercopithecus* **	34.59 (7)	3.41	45.77(7)	3.58	36.00 (3)	10.11
** *cf. Chlorocebus* **	30.61 (48)	2.43	38.59 (45)	4.01	30.83 (37)	3.77
** *Colobus* **	43.59 (85)	5.23	51.64 (73)	7.44	48.50 (59)	6.38
** *Kuseracolobus* **	95.88 (1)	-	103.4 (1)	-	76.54 (1)	-
** *Libypithecus* **	48.90 (2)	2.97	58.93 (2)	3.78	65.95 (2)	0.49
** *Papio* **	95.81 (37)	13.39	139.68 (42)	22.23	137.82 (33)	29.59
** *Paracolobus* **	84.95 (4)	27.92	127.95 (9)	31.10	129.37 (8)	39.24
** *Parapapio* **	97.06 (20)	16.92	135.16 (20)	24.73	127.91 (28)	23.93
** *Pliopapio* **	54.02 (1)	-	73.1 (1)	-	66.75 (1)	-
** *Procercocebus* **	97.83 (4)	7.80	134.44 (5)	10.40	120.29 (5)	11.99
** *Rhinocolobus* **	85.70 (4)	8.13	104.31 (3)	2.92	114.32 (4)	13.56
** *Soromandrillus* **	121.41 (4)	12.21	198.79 (6)	28.07	205.49 (6)	30.60
** *Theropithecus* **	1.34 (6)	0.03	1.34 (6)	0.03	1.34 (6)	0.03
** *Victoriapithecus* **	42.19 (8)	1.30	57.45 (9)	4.58	44.55 (5)	4.15

* M1, M2, and M3 areas refer to the two-dimensional area of the tooth in occlusal view, calculated as the mesiodistal length multiplied by the buccolingual breadth. See text for details and definitions. StDv = standard deviation. See Appendix A for more extensive descriptive statistics.

**Table 5 biology-11-01218-t005:** Descriptive statistics for the Genotype:Phenotype (G:P)-mapped traits *.

	MMC		PMM		IC	
	Mean (n)	StDv	Mean (n)	StDv	Mean (n)	StDv
**Extant Genera**						
** *Cercocebus* **	1.05 (15)	0.05	1.42 (15)	0.06	1.10 (15)	0.10
** *Cercopithecus* **	0.95 (92)	0.06	1.49 (103	0.08	0.97 (76)	0.09
** *Chlorocebus* **	0.96 (24)	0.07	1.48 (26)	0.10	1.02 (24)	0.13
** *Colobus* **	1.08 (118)	0.06	1.47 (120)	0.07	1.15 (93)	0.10
** *Erythrocebus* **	1.04 (1)	-	1.48 (1)	-	1.20 (1)	-
** *Lophocebus* **	0.98 (5)	0.05	1.49 (6)	0.03	1.04 (5)	0.11
** *Macaca* **	1.12 (149)	0.08	1.59 (174)	0.09	1.26 (148)	0.14
** *Mandrillus* **	1.16 (18)	0.05	1.45 (18)	0.08	1.38 (18)	0.12
** *Nasalis* **	1.09 (30)	0.05	1.57 (29)	0.08	1.12 (30)	0.10
** *Papio* **	1.22 (86)	0.07	1.65 (99)	0.09	1.46 (71)	0.13
** *Piliocolobus* **	1.08 (15)	0.05	1.47 (15)	0.04	1.15 (15)	0.09
** *Presbytis* **	0.97 (151)	0.05	1.41 (160)	0.07	0.96 (138)	0.08
** *Theropithecus* **	1.25 (8)	0.05	1.88 (10)	0.06	1.53 (8)	0.06
**Fossil Genera**						
** *Cercopithecoides* **	1.18 (13)	0.15	1.64 (18)	0.15	1.27 (7)	0.25
** *Cercopithecus* **	0.92 (2)	0.03	1.51 (4)	0.15	0.85 (2)	0.04
** *cf. Chlorocebus* **	0.98(26)	0.07	1.55 (40)	0.11	1.01 (22)	0.09
** *Colobus* **	1.07 (55)	0.09	1.44 (66)	0.08	1.12 (46)	0.10
** *Kuseracolobus* **	1.17 (1)	-	1.65 (2)	0.14	-	-
** *Libypithecus* **	1.14 (2)	0.06	1.41 (2)	0.10	1.35 (2)	0.09
** *Papio* **	1.20 (32)	0.10	1.72 (36)	0.14	1.42 (21)	0.24
** *Paracolobus* **	1.16 (3)	0.20	1.54 (4)	0.10	1.26 (3)	0.23
** *Parapapio* **	1.18 (23)	0.13	1.74 (22)	0.11	1.34 (19)	0.21
** *Pliopapio* **	1.20 (1)	-	1.72 (1)	-	1.24 (1)	-
** *Procercocebus* **	1.13 (4)	0.03	1.61 (5)	0.03	1.42 (2)	0.19
** *Rhinocolobus* **	1.19 (3)	0.07	1.43 (3)	0.07	1.26 (2)	0.16
** *Soromandrillus* **	1.28 (6)	0.05	1.78 (6)	0.13	1.55 (4)	0.10
** *Theropithecus* **	1.34 (6)	0.03	1.34 (6)	0.03	1.62 (5)	0.06
** *Victoriapithecus* **	1.01 (4)	0.05	1.59 (2)	0.08	1.05 (3)	0.12

* MMC = molar module component; PMM = premolar-molar module; IC = inhibitory cascade. See text for details and definitions. StDv = standard deviation. See Appendix A for more extensive descriptive statistics.

**Table 6 biology-11-01218-t006:** Phylogenetic ANOVA results for extant genera *.

		Traits					
		M1 2D Area	M2 2D Area	M3 2D Area	IC	MMC	PMM
Summary *p*-Value		0.0004	<0.0001	<0.0001	0.003	0.004	0.009
	**Genera:**						
Colobines	*Presbytis*	**0.027 ***	**0.004 ****	**0.003 ****	0.078	0.100	0.765
	*Nasalis*	0.560	0.381	0.436	0.967	0.385	**0.029 ***
	*Piliocolobus*	0.153	**0.041 ***	0.091	0.663	0.503	0.270
	*Colobus*	0.346	0.206	0.277	0.659	0.510	0.252
Papionins	*Macaca*	0.233	0.310	0.474	0.075	0.088	**0.006 ****
	*Papio*	**<0.0001 *****	**<0.0001 *****	**<0.0001 *****	**0.007 ****	**0.009 ****	**0.005 ****
	*Theropithecus*	**0.006 ****	**0.0002 *****	**<0.0001 *****	**0.003 ****	**0.005 ****	**0.0002 *****
	*Lophocebus*	0.380	0.227	0.123	0.361	0.251	0.112
	*Mandrillus*	**0.008 ****	**0.0003 *****	**<0.0001 *****	**0.002 ****	**0.004 ****	0.288
Cercopith-ecins	*Chlorocebus*	0.077	**0.029 ***	**0.017 ***	0.308	0.111	0.238
	*Erythrocebus*	0.151	0.068	0.123	0.371	0.899	0.174
	*Cercopithecus*	**0.019 ***	**0.004 ****	**0.002 ****	**0.039 ***	**0.026 ***	0.091

* M1, M2, M3 refer to the first, second, and third molars. 2D refers to the two-dimensional area of the tooth crown in occlusal view, calculated as the mesiodistal length multiplied by the buccolingual breadth. IC is the 2-d area of the M3 divided by the 2D area of the M1. MMC is the mesiodistal length of the M3 divided by the mesiodistal length of the M1. PMM is the mesiodistal length of the M2 divided by the mesiodistal length of the P4 (fourth premolar). All area traits were geometric mean size-corrected before analysis. * indicates significance at *p* < 0.05. ** indicates significance at *p* < 0.01. *** indicates significance at *p* < 0.001.

**Table 7 biology-11-01218-t007:** Blomberg’s K for the dental traits *.

Trait	K-Value	K *p*-Value
M1A	0.6595	0.070
M2A	0.6727	0.058
M3A	0.6606	0.055
**IC**	**0.6251**	**0.045**
**MMC**	**0.6324**	**0.035**
PMM	0.6379	0.059

* M1A, M2A, and M3A = two-dimensional area estimates for first, second, and third maxillary molars; MMC = molar module component; PMM = premolar-molar module; IC = inhibitory cascade. See text for trait definitions. Statistically significant estimates are in bold text. K-values greater than 1 indicate a strong phylogenetic signal. Non-significant *p*-values are interpreted as evolution under neutral genetic drift. For K-values that are significant at *p* < 0.05, the trait is interpreted to show evidence of selection.

**Table 8 biology-11-01218-t008:** Comparison of trait values from the Ancestral State Reconstruction (ASR) and Possible Fossil Representatives *.

ASR Node	ASR MMC	ASR PMM	Molecular Divergence	Possible Fossil Representative	MMC Value	PMM Value	Geological Age
20	1.07	1.52	16 Ma	*Victoriapithecus*	1.03	1.59	19–12.5 Ma
28	1.14	1.45	5 Ma	*Procercocebus*	1.13	1.62	2.5 Ma
28	1.14	1.45	5 Ma	*Soromandrillus*	1.28	1.78	2–3 Ma
29	1.10	1.43	2 Ma	*Procercocebus*	1.13	1.62	2.5 Ma
30	1.18	1.45	2.5 Ma	*Soromandrillus*	1.28	1.78	2–3 Ma
31	1.17	1.66	2 Ma	*Parapapio*	1.18	1.76	2–5 Ma
31	1.17	1.66	2 Ma	*Pliopapio*	1.20	1.72	4.4 Ma
35	1.07	1.49	<7.5 Ma	*Paracolobus*	1.16	1.50	2–6 Ma
35	1.07	1.49	<7.5 Ma	*Cercopithecoides*	1.18	1.67	2–5 Ma
35	1.07	1.49	<7.5 Ma	*Kuseracolobus*	1.17	1.75	4–4.4 Ma
35	1.07	1.49	<7.5 Ma	*Libypithecus*	1.14	1.41	5 Ma
**ASR Tip**	**MMC**	**PMM**	**Molecular** **Divergence**	**Possible Fossil** **Representative**	**MMC Value**	**PMM Value**	**Geological Age**
*Chlorocebus aethiops*	0.96	1.48	1 Ma	cf. *Chlorocebus* (Ethiopia)	0.98	1.56	100–600 ka
*Colobus guereza*	1.08	1.47	<1.6 Ma	*Colobus* sp. (Ethiopia)	1.06	1.44	100–600 ka

* Ma = million years ago; ka = thousand years ago; MMC and PMM are defined in the text; Molecular divergence estimates: Node 20 [92]; Node 28–31 [93]; Node 35 [94]. Geological dates for the fossils: *Victoriapithecus*, *Parapapio*, *Paracolobus*, *Cercopithecoides* [95]; *Procercocebus* [96]; *Soromandrillus* [97]; *Pliopapio* [66]; *Kuseracolobus* [73]; *Libypithecus* [98]; cf. *Chlorocebus* (authors, unpublished data); *Colobus* (authors, unpublished data).

## Data Availability

Data are available through the publications cited herein and the published dataset [65].

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
