# Peer review of "Keeping 21st Century Paleontology Grounded: Quantitative Genetic Analyses and Ancestral State Reconstruction Re-Emphasize the Essentiality of Fossils"

_biology, 2022, doi:10.3390/biology11081218_

Round 1

Reviewer 1 Report

This is a nice example of what genotype:phenotype mapping offers for analyses of primate evolution and diversity. The authors study traditional molar size metrics and three traits informed by quantitative genetics or developmental biology: Mandibular Molar Components (MMC), Premolar-Molar Components (PMC), and Inhibitory Cascade (IC) Model. These three metrics are essentially ratios between dental metrics that reflect the shared genetic or developmental patterns or constraints between the molar/pre-molar teeth. From these data they test two hypotheses: 1) G:P mapped traits show evidence of ancestry and selection useful for paleontological investigations and 2) that G:P mapped traits have a range of variation that is not predicted based on extant variation alone. 

The Introduction is clearly presented with lucid descriptions of the different traits and explicit statements of hypotheses. However, could the relationship between body size and molar dimensions be fleshed out a bit more for this analysis. It is stated on lines 128-129 that molar width is genetically correlated with body size, but molar length is not (emphasis added). Please clarify what it means to be genetically correlated in this case. Please expound a bit on the lack of a correlation for molar length. This will be useful for interpreting the ANOVA results below.  

The Materials and Methods are also generally clear, however the description of Blomberg’s K could be improved, particularly the relative implications of K value > 1 vs. K value < 1.

Results

Heritability Tests

The presentation of these results is straightforward. It is prudent to take the conservative approach with respect to non-significant results reflecting low sample size. Is the H2 value for L MMC at 0.001 correct? If so, this seems to be quite an outlier compared to all the other results being significant or nearly so with a non-zero value. It also seems weird that a H2 value of 0.001 would have even a non-significant p-value as low as 0.496. 

Phylogenetic ANOVA

Tie in the previous finding of the different relationships of molar width and length to body size here. While I don’t implicitly doubt your findings, for those of us used to studying postcrania, it is difficult comprehend the comparison of any morphological traits that do not initially control for the effects of size. Please discuss this issue here for the area metrics. 

Phylogenetic Signal

The K values are generally similar for all metrics or indices, however only a few pass the significance threshold. When discussing non-significant heritability above, low sample size was discussed as a potential factor. How does sample size affect these results? I am not advocating for over-interpreting the results, however I have seen in previous analysis that the null-hypothesis of neutral evolution can be difficult to reject making these tests very conservative. 

ASR analysis and discussion are fantastic. 

Reviewer 2 Report

Dear editor and authors. The manuscript is excellent. The arguments and analyzes are very good and really manage to test the hypotheses proposed in the study. The discussion is very in-depth and interesting and will certainly inspire other authors to carry out similar studies in other groups of mammals. However, there are small issues to be corrected, which I highlight the absence of italics in most scientific names in the body of the manuscript text. I have marked my suggestions in the attached pdf file. The issues raised by me do not detract from the manuscript and, therefore, I consider that the manuscript should be published after the authors make the small changes I have indicated.

Author Response

Many thanks to the Reviewer for the positive feedback and suggestions for improving the text. We have corrected all the missing italics for the Linnaean names (these were lost in the conversion to the MDPI format during submission). We have also corrected the year for the Wilson et al. reference that we mixed up (thank you for catching that!). We appreciate the suggestion to remove the text in the paragraph at lines 183 to 203 that reviews our 2016 publication in order to streamline the text. We would prefer to keep this section in this manuscript in order to make it easier for readers who are unfamiliar with that previous work and may not have time to go read that other paper. 

Thank you for your help!